# Series-Parallel Reconfiguration Technique with Voltage Equalization Capability for Electric Double-Layer Capacitor Modules

**Masatoshi Uno [1],\* , Koyo Iwasaki [2] and Koki Hasegawa [3]**

1    College of Engineering, Ibaraki University, Hitachi 316-8511, Japan
2    Fanuc Corporation, Yamanashi 401-0597, Japan
3    Seiko Epson Corporation, Nagano 392-8502, Japan
\*    Correspondence: masatoshi.uno.ee@vc.ibaraki.ac.jp

**Abstract:** Voltage variations of electric double-layer capacitors (EDLCs) are rather wider than those of traditional rechargeable batteries, and an energy utilization ratio of EDLCs is dependent on cells' voltage variation ranges. To satisfactorily utilize EDLCs' energies, voltages of EDLC modules should be within a certain range, while cells need to be charged and discharged over the wide voltage range. To this end, various kinds of series-parallel reconfiguration techniques based on balance- and unbalance-shift circuits have been proposed, but conventional techniques can only be applied to modules consisting of even number cells, impairing the design flexibility and scalability. With the unbalance-shift circuits, cell voltages are unavoidably mismatched due to unequal currents, resulting in reduced energy utilization ratios. This article proposes a novel series-parallel reconfiguration technique with voltage equalization capability for EDLC modules. The proposed technique can be applied to any number of cells, improving design flexibility and scalability. Furthermore, since the proposed circuit behaves as a switched capacitor converter, in which all cells are virtually connected in parallel, cells are equally charged and discharged without causing voltage imbalance, realizing the improved energy utilization ratio. A prototype for an EDLC module comprising four cells, each with a rated charging voltage of 2.5 V, was built and experimentally tested. The module voltage varied in the range of 3.2–5.0 V, while all cells were uniformly discharged down to as low as 0.8 V, achieving the energy utilization ratio of 90%.

**Keywords:** electric double-layer capacitor (EDLC); series-parallel reconfiguration; switched capacitor converter; voltage equalization

## 1. Introduction

Electric double-layer capacitors (EDLCs), also known as supercapacitors, are an energy storage device that utilizes double-layer capacitance as an energy storage mechanism. EDLCs excel over traditional rechargeable batteries in terms of power capability, cycle life, and temperature tolerance because they do not rely on chemical reactions for energy storage [1,2]. Their major drawback is that their specific energy in W/kg and energy density in W/L are rather smaller than those of rechargeable batteries. Hence, EDLCs have been chiefly used as high-power energy buffers to supplement main power sources, such as lithium-ion batteries (LIBs) and fuel cells. Lithium-ion capacitors (LICs)—a hybrid energy storage device combining the energy storage mechanisms of EDLCs and LIBs—realize higher specific energy and energy density than EDLCs [3–9]. EDLCs and LICs have been commercialized by various manufacturers, and some applications have started employing EDCLs and LICs as an alternative to traditional rechargeable batteries in order to achieve a longer

service life and wider operation temperature range. In spacecraft power systems, for example, a feasibility study on LIC-based power systems was performed [10]. A LIC pouch cell was launched and its performance under a space environment was successfully demonstrated on orbit [9]. Another example is an uninterruptible power supply (UPS) application, where maintenance-free rechargeable energy storage devices are strongly demanded, and EDLCs have been recently used as an alternative to traditional lead-acid batteries which typically need replacement every few years.

In addition to the lower energy densities, a wide voltage variation of EDLCs during cycling is often cited as a major disadvantage. Typical voltage ranges of EDLCs are 0–2.5 V, whereas those of LIBs and LICs are typically 2.7–4.2 V and 2.2–3.8 V, respectively. Typical discharging voltage profiles of an EDLC, LIB, and LIC cells as a function of ampere-hour depth-of-discharge (DoD) are compared in Figure 1.

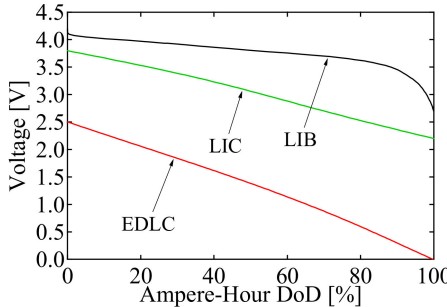

**Figure 1.** Voltage variations of lithium-ion battery (LIB), lithium-ion capacitor (LIC), and electric double-layer capacitor (EDLC) cells as a function of ampere-hour DoD.

In general, an energy utilization ratio of EDLCs, *U*, is expressed as:

$$U = \frac{V_{cha}{}^2 - V_{cut}{}^2}{V_{cha}{}^2} = 1 - \left(\frac{V_{cut}}{V_{cha}}\right)^2 \tag{1}$$

where $V_{cha}$ and $V_{cut}$ are the charge and cut-off voltages, respectively. Figure 2 illustrates *U* as a function of $V_{cut}/V_{cha}$. This relationship suggests that to enhance energy utilization, EDLCs need to be discharged deeply so that $V_{cut}/V_{cha}$ is small. For examples, if EDLCs are discharged to 0.5 $V_{cha}$, 75% of the stored energy can be used. To achieve 90% energy utilization, EDLCs need to be discharged to 0.32 $V_{cha}$. However, the range of $V_{cut}/V_{cha}$ is typically constrained by dc–dc converters that are connected in between EDLCs and loads. Poor utilization ratios are equivalent to decreased specific energies and energy densities because a certain amount of energy stored in EDLCs is unavailable and unextractable, exacerbating the drawback of EDLCs.

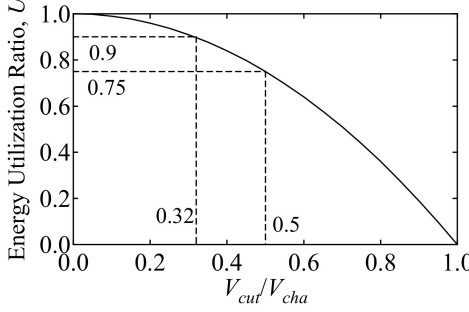

**Figure 2.** Energy utilization ratio of EDLC as a function of ratio of cut-off voltage to charge voltage ($V_{cut}/V_{cha}$).

Various kinds of series-parallel reconfiguration techniques, or changeover circuits, have been developed to address the above issues [11–17]. EDLC cells together with bidirectional switches form

a reconfiguration module so that series and parallel connections of cells are switched depending on cell voltages. Reconfiguration techniques can be roughly categorized into two groups: balance- and unbalance-shift circuits.

A typical reconfiguration sequence of the balance-shift circuit for four-cell module is illustrated in Figure 3a, and its discharging and charging profiles are shown in Figure 3b,c, respectively. At the beginning of discharging, all cells are connected in parallel, and hence, the module is equivalent to 1-series 4-parallel configuration (i.e., 1S-4P). After cell voltages drop to some extent, the module is reconfigured to be 2-series–2-parallel configuration (2S-2P) to double its voltage. After further discharging, all cells are connected in series (4S-1P) to lift up the module voltage again. With this reconfiguration sequence, cells can deeply discharge while the module voltage variation can be relatively small. In the charging process, the series-parallel combination is sequentially changed in reverse order. With the balance-shift circuit, all EDLC cells can be charged and discharged uniformly without producing voltage imbalance, as illustrated in the bottom panels of Figure 3b,c.

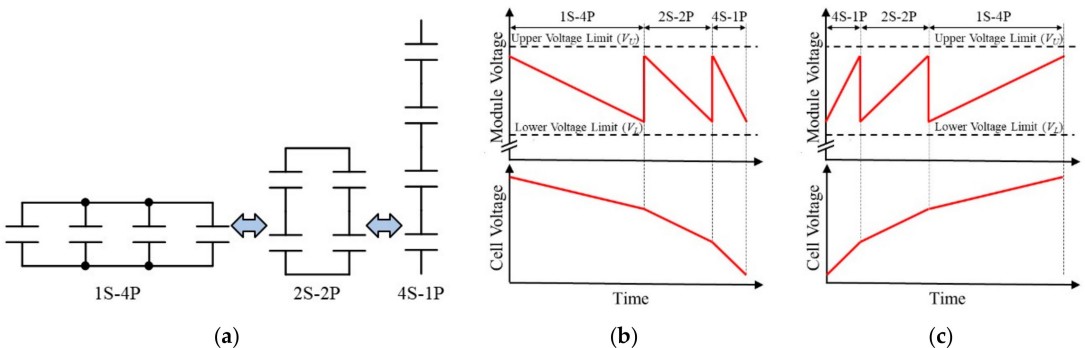

**Figure 3.** (**a**) Typical reconfiguration sequence of balance-shift circuit for four-cell module, (**b**) its discharging, and (**c**) charging profiles.

Various kinds of balance-shift circuits have been proposed [11–16], as shown in Figures 4 and 5. Cells in these balance-shift circuits can be reconfigured identically to those in Figure 3, and their configurations are either 1S-4P, 2S-2P, or 4S-1P, depending on cell voltages. Although all cells can be charged and discharged uniformly with a high utilization ratio, the requirement of numerous switches is cited as a top concern. Furthermore, these techniques cannot be applied to modules consisting of odd number cells (e.g., 3, 5, and 7 cells), impairing the design flexibility and scalability.

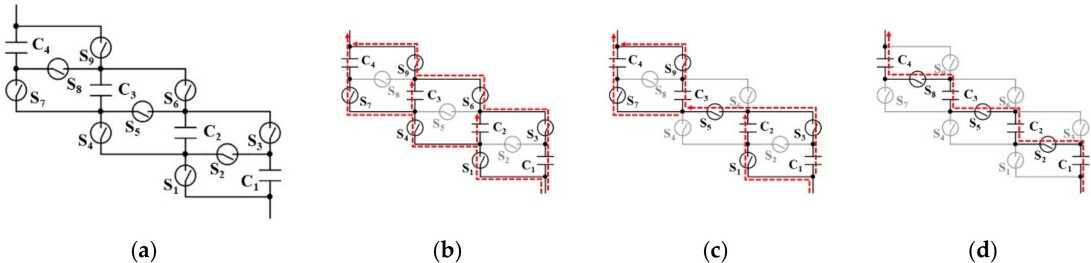

**Figure 4.** (**a**) Balance-shift circuit, (**b**) 1-series 4-parallel (1S-4P), (**c**) 2S-SP, and (**d**) 4S-1P.

A representative reconfiguration sequence of the unbalance-shift circuit for four cells are depicted in Figure 6a, and discharging and charging profiles of module and cell voltages are shown in Figure 6b,c, respectively. The number of series connection can be changed step by step (i.e., 2S, 3S, and 4S), realizing finer voltage step changes than balance-shift circuits. However, its major drawback is that cells are unequally charged and discharged during intermediate mode(s). In a discharging process, for example, all cells share the same discharge current in 2S-2P and 4S-1P configurations in the first and third modes, but cells in the second mode are obviously discharged unequally. This unequal current impairs the

energy utilization ratio of the module as a whole because some cells might reach 0 V or, in the worst case, be over-discharged under subzero voltages.

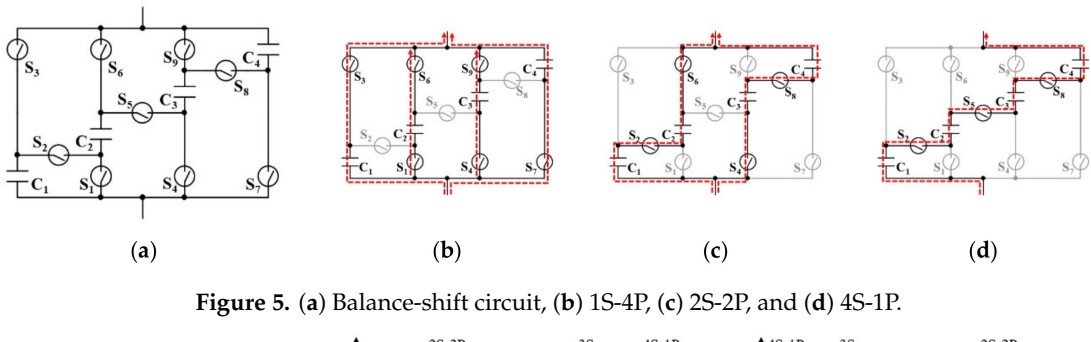

**Figure 5.** (**a**) Balance-shift circuit, (**b**) 1S-4P, (**c**) 2S-2P, and (**d**) 4S-1P.

**Figure 6.** (**a**) Typical reconfiguration sequence of unbalance-shift circuit for four-cell module, (**b**) its discharging, and (**c**) charging profiles.

Unbalance-shift circuits for four cells [17] and 2 *m* cells [16] are shown in Figure 7a,b, respectively. Reconfiguration modes during discharging are shown in Figure 7c–e. In comparison to the balance-shift circuits shown in Figures 4 and 5, the switch count can be reduced, thus simplifying the circuit. However, cell voltages are naturally imbalanced due to the unequal current in the intermediate mode, as depicted in Figure 7d.

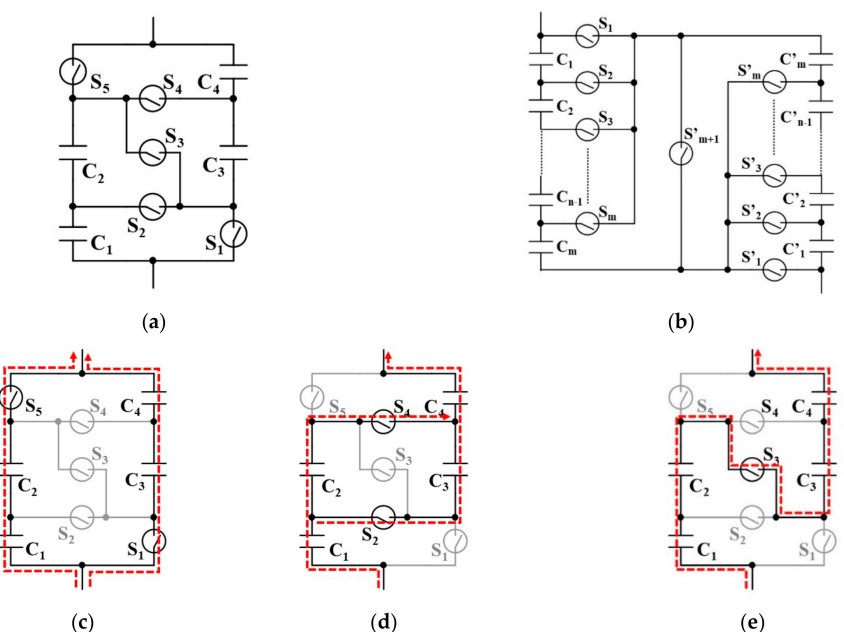

**Figure 7.** Unbalance-shift circuit for (**a**) four cells and (**b**) 2 *m* cells. Operation modes of unbalance-shift circuit for four cells; (**c**) 2S-2P, (**d**) 3S, and (**e**) 4S-1P.

To cope with the issues of conventional reconfiguration techniques, this article presents a novel reconfiguration circuit with voltage equalization capability. The proposed reconfiguration circuit can be applied to modules consisting of any arbitrary odd and even numbered cells, realizing the improved design flexibility and scalability. In addition, cells in the proposed reconfiguration circuit can be charged and discharged without causing voltage imbalance thanks to the voltage equalization capability, hence achieving the improved energy utilization ratios. The rest of this article is organized as follows. Section 2 describes the proposed reconfiguration circuit and its major features. The operation analysis is performed in Section 3, followed by the design example in Section 4. Section 5 presents the experimental results of an EDLC module consisting of four cells.

## 2. Series-Parallel Reconfiguration Circuit with Voltage Equalization Capability

### 2.1. Circuit Description

The proposed reconfiguration circuits for three cells and four cells are shown in Figure 8a,b, respectively. Series and parallel connections of cells are reconfigured by switches. Diodes are connected in parallel with switches in order to provide current flow paths during dead time periods (detailed in Section 3).

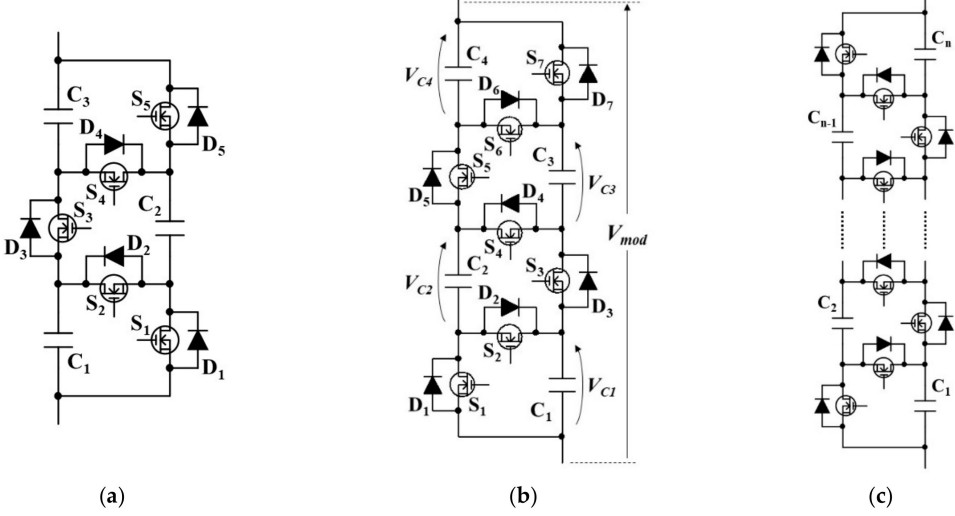

|  (**a**)  |  (**b**)  |  (**c**)  |

**Figure 8.** Proposed series-parallel reconfiguration module for (**a**) three cells, (**b**) four cells, and (**c**) *n* cells.

The proposed reconfiguration technique can be applied to any arbitrary odd and even numbered cells. The proposed reconfiguration circuit for *n* cells is depicted in Figure 8c. The number of cells can be arbitrarily extended by adding two switches for every cell. The cell count *n* can be any integer, including odd numbers.

The configuration of the circuit for three cells (Figure 8a) in a discharging process is switched in the order of 2S and 3S-1P. All cells uniformly discharge in series in the 3S-1P configuration, and therefore no voltage imbalance occurs. In the 2S configuration, on the other hand, cells are discharged with different current rates and their voltages tend to be imbalanced. To equalize the cell voltages, the series-parallel connection of these three cells are sequentially changed. Similarly, the series-parallel configuration of the module for four cells (Figure 8b) in a discharging process is changed in the order of 2S-2P, 3S, and 4S-1P. Cells uniformly discharge in the 2S-2P and 4S-1P configurations, whereas those in the 3S configuration are discharged unequally. Therefore, the series-parallel connections of four cells are repeatedly changed in order to preclude voltage imbalance in the 3S configuration. In Section 3, operations of a module for four cells in discharging and charging processes is explained.

## 2.2. Major Features

As mentioned in Section 2.1, the proposed reconfiguration circuit can be applied to any number of cells, hence improving the design flexibility and scalability in comparison to conventional circuits, which can only be applied to even numbered cells, as mentioned in Section 1. In addition, the switch count can be reduced in comparison with balance-shift circuits, realizing a simplified circuit.

As will be detailed in Section 3, the proposed reconfiguration circuit behaves as a switched capacitor converter [18–21], in which all cells are virtually connected in parallel and their voltages are automatically balanced, and therefore, all cells can be charged and discharged uniformly. In other words, all cells are utilized equally, realizing an improved utilization ratio of the module compared to the conventional unbalance-shift circuit shown in Figure 7.

The proposed series-parallel reconfiguration circuit is compared to conventional balance- and unbalance-shift circuits in terms of the switch count, the applicable number of cells, and voltage balance capability, as shown in Table 1 where $n$ ($\geq 2$) is the number of cells in a module. Cells in the balance-shift circuit can be charged and discharged equally, but its switch count is prone to be larger than other circuits. The unbalance-shift circuit requires the fewest switch count, but it suffers from voltage imbalance, as illustrated in Figure 6. Both conventional circuits are applicable to only even numbered cells (e.g., 4, 6, 8 . . . ), hence impairing the design flexibility and scalability. Meanwhile, the proposed reconfiguration circuit can be applied to any number of cells, and therefore, the modules can be flexibly designed and scaled up to be larger systems. Furthermore, all cells in the proposed reconfiguration circuit can be charged and discharged equally without causing voltage imbalance, achieving higher energy utilization than the conventional unbalance-shift circuit.

**Table 1.** Comparison among conventional and proposed reconfiguration circuits.

| Topology | Balance-Shift Circuit | Unbalance-Shift Circuit | Proposed |
|---|---|---|---|
| Switch Count | $3(n-1)$ | $n+1$ | $2n-1$ |
| Applicable Cell Number | Even Number | Even Number | Any Number Cells |
| Voltage Balance | Balanced | Unbalanced | Balanced |

According to previous work on reconfiguration circuits [11,16], the larger the number of stages or cells $n$, the better will be the energy utilization ratio $U$. However, the circuit complexity is prone to soar due to the increased switch count. Therefore, reconfiguration techniques have generally been employed for modules consisting of around four cells—for large-scale systems, reconfiguration techniques are applied at a module-level, not a cell-level (i.e., $C_1$–$C_4$ in figures in Section 1 are a module, not a cell). This tendency is also true for the proposed reconfiguration technique, and therefore, the circuit for four cells is explained and discussed in the following sections.

## 3. Operation Analysis

Depending on cell voltages, the series-parallel configuration of the proposed circuit is sequentially changed, similarly to conventional reconfiguration circuits. However, in intermediate modes, the proposed circuit behaves as a switched capacitor converter [18–21], and all cell voltages are balanced, realizing uniform cell utilization. This section deals with the circuit for four cells, but operations of a circuit for $n$ cells can also be explained similarly.

### 3.1. Voltage Profiles and Mode Transitions

Theoretical voltage profiles in discharging and charging modes are shown in Figure 9a,b, respectively. The proposed reconfiguration circuit for four cells operates chiefly with three modes, Modes 1–3, and Mode 2 consists of three sub-modes, sub-Modes A–C. These modes and sub-modes are sequentially shifted with inserting a dead time period in order to prevent excessive large currents due to possible short-circuit paths.

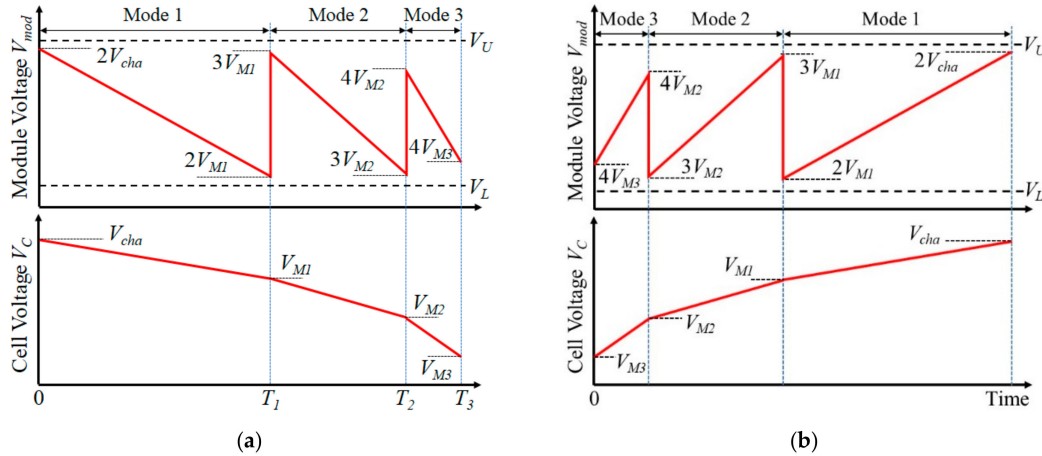

**Figure 9.** Module voltage $V_{mod}$ and cell voltage $V_C$ in (**a**) discharging and (**b**) charging.

The mode transition map is depicted in Figure 10. In a discharging process, the proposed reconfiguration circuit operates in the order of Mode 1, 2, and 3. At a moment of the mode transition from Mode 1 to 2, any sub-modes in Mode 2 can be directly reached from Mode 1 via a dead time period. In Mode 2, sub-Modes A–C are periodically switched in order to prevent voltage imbalance or to equalize cell voltages. At a moment of the mode transition from Mode 2 to 3, the operation can shift to Mode 3 from any sub-modes in Mode 2. In a charging process, on the other hand, the operation shifts in the order of Mode 3, 2, and 1, but the fundamental operating principle is identical to that in the discharging process.

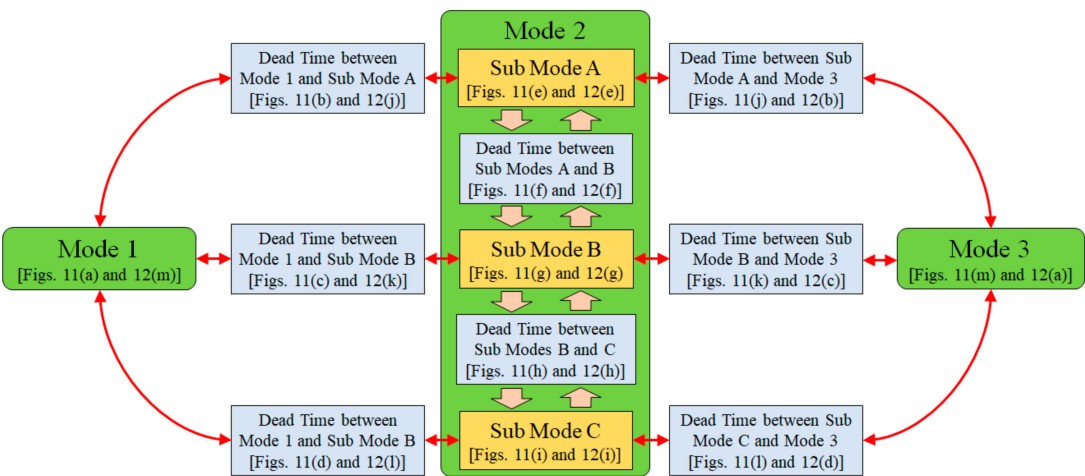

**Figure 10.** Mode transition map.

### 3.2. Operation in Discharging Mode

Voltage profiles and operation modes in the discharging process are shown in Figures 9a and 11, respectively. As described in the previous subsection and in Figure 10, there are three ways to/from sub-modes in Mode 2. In this subsection, the mode transitions from Mode 1 to sub-Mode A and from sub-Mode C to Mode 3 are taken as example cases to be explained.

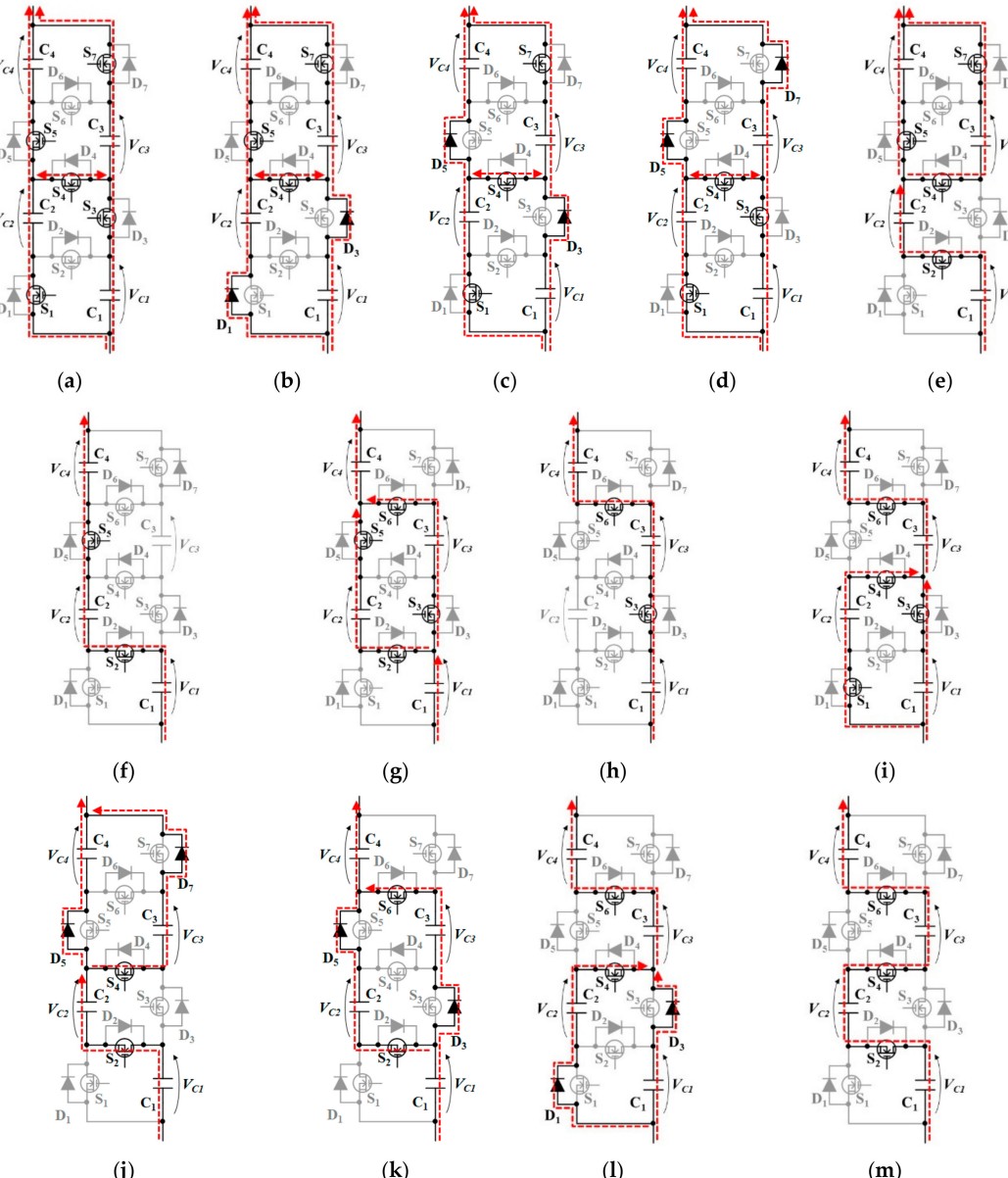

**Figure 11.** Mode transitions in the discharging process: Current flow paths in (**a**) Mode 1, (**b**) dead time between Mode 1 and sub-Mode A, (**c**) dead time between Mode 1 and sub-Mode B, (**d**) dead time between Mode 1 and sub-Mode C, (**e**) sub-Mode A, (**f**) dead time between sub-Modes A and B, (**g**) sub-Mode B, (**h**) dead time between sub-Modes B and C, (**i**) Mode C, (**j**) dead time between sub-Mode A and Mode 3, (**k**) dead time between sub-Mode B and Mode 3, (**l**) dead time between sub-Mode C and Mode 3, and (**m**) Mode 3.

Mode 1 (Figure 11a): $S_1$, $S_3$, $S_4$, $S_5$, and $S_7$ are on. The reconfiguration circuit is equivalent to 2S-2P. $C_1$ and $C_2$ are connected in parallel, and so are $C_3$ and $C_4$. The module voltage, $V_{mod}$, at the beginning of this mode is equal to $2V_{cha}$ ($V_{cha}$ being the charge voltage of cells). Voltages of parallel-connected cells are naturally balanced. $V_{mod}$ as well as cell voltages $V_C$ decreases, and all cells are equally discharged unless the total capacitance of $C_1$ and $C_2$ severely differs from that of $C_3$ and $C_4$. At the end of Mode 1, $V_C$ and $V_{mod}$ decrease to $V_{M1}$ and $2V_{M1}$ ($V_{M1}$ being the cell voltage at the end of Mode 1), respectively.

Dead Time between Mode 1 and Sub-Mode A (Figure 11b): $S_1$ and $S_3$ are turned off, and $D_1$ and $D_3$ start conducting. Meanwhile, $S_4$, $S_5$, and $S_7$ are still on. The module configuration is still 2S-2P.

Sub-Mode A in Mode 2 (Figure 11e): $S_4$, $S_5$, and $S_7$ are still on, whereas $S_2$ is turned on so as to reconfigure the circuit to be 3S. $C_1$ and $C_2$ are in series through $S_2$, and hence, the module voltage $V_{mod}$

jumps to as high as $3V_{M1}$ at the beginning of Mode 2. $C_3$ and $C_4$ are connected in parallel, and their voltages are balanced.

Dead Time between Sub-Modes A and B (Figure 11f): $S_4$ and $S_7$ are turned off while $S_2$ and $S_5$ are still on. Although $C_3$ does not contribute to discharging, the module configuration is still 3S.

Sub-Mode B in Mode 2 (Figure 11g): $S_3$ and $S_6$ are turned on, and $C_2$ and $C_3$ are connected in parallel. The module configuration is 3S, and voltages of $C_2$ and $C_3$ are naturally balanced in this sub-mode.

Dead Time between Sub-Modes B and C (Figure 11h): $S_3$ and $S_6$ are kept on, while $S_2$ and $S_5$ are turned off. Although $C_2$ does not contribute to discharging, the module configuration is still 3S.

Sub-Mode C in Mode 2 (Figure 11i): $S_1$ and $S_4$ are turned on so as to connect $C_1$ and $C_2$ in parallel. The module configuration is 3S, and voltages of $C_1$ and $C_2$ are naturally balanced.

By repeating sub-Modes A–C with inserting dead time periods, all cells are virtually connected in parallel and their voltages are automatically balanced in Mode 2. Meanwhile, the module configuration can be maintained to be 3S, even though series-parallel connections are sequentially changed. At the end of Mode 2, $V_C$ and $V_{mod}$ decrease to $V_{M2}$ and $3V_{M2}$ ($V_{M2}$ being the cell voltage at the end of Mode 2), respectively.

Dead Time between Sub-Mode C and Mode 3 (Figure 11l): $S_1$ and $S_3$ are turned off, and $D_1$ and $D_3$ conduct. Meanwhile, $S_4$ and $S_6$ are still on. The configuration is still 3S.

Mode 3 (Figure 11m): $S_2$ is turned on, and all cells are connected in series, forming the 4S-1P configuration. $V_{mod}$ jumps to $4V_{M2}$. All cells uniformly discharge in series, and no voltage imbalance occurs unless capacitances are severely mismatched. $V_C$ declines to $V_{M3}$ at the end of Mode 3.

In summary, module configurations in Modes 1 and 3 are identical to those in the conventional circuits shown in Figure 7. In Mode 2, all cells are virtually connected in parallel thanks to the switched capacitor operation throughout sub-Modes A–C.

### 3.3. Operation in Charging Mode

Voltage profiles and operation modes in the charging process are shown in Figures 9b and 12, respectively. The proposed reconfiguration circuit in the charging process operates in a similar manner to the discharging process. In this subsection, mode transitions from Mode 3 to sub-Mode C and from sub-Mode A to Mode 1 are discussed.

Mode 3 (Figure 12a): $S_2$, $S_4$, and $S_6$ are on, and the circuit is equivalent to 4S-1P. All cells are charged uniformly as long as cell capacitances are matched satisfactorily. $V_C$ and $V_{mod}$ reach $V_{M2}$ and $4V_{M2}$ at the end of Mode 3.

Dead Time between Mode 3 and Sub-Mode C (Figure 12d): $S_2$ is turned off, and its anti-parallel diode $D_2$ conducts instead. The configuration is still 4S-1P.

Sub-Mode C in Mode 2 (Figure 12i): $S_1$ and $S_3$ are turned on to connect $C_1$ and $C_2$ in parallel. The module configuration is 3S, and therefore, $V_{mod}$ falls down to $3V_{M2}$. Voltages of $C_1$ and $C_2$ are balanced as these cells are connected in parallel.

Dead Time between Sub-Modes C and B (Figure 12h): $S_3$ and $S_6$ are still on, whereas $S_1$ and $S_4$ are turned off. Although $C_2$ is not being charged in this sub-mode, the module is still 3S configuration.

Sub-Mode B in Mode 2 (Figure 12g): $S_2$ and $S_5$ are turned on while $S_3$ and $S_6$ are kept on. $C_2$ and $C_3$ are connected in parallel, and their voltages are unified. The module configuration is still 3S.

Dead Time between Sub-Modes B and A (Figure 12f): $S_3$ and $S_6$ are turned off. Although $C_3$ is isolated from the charging path, the module configuration is still 3S.

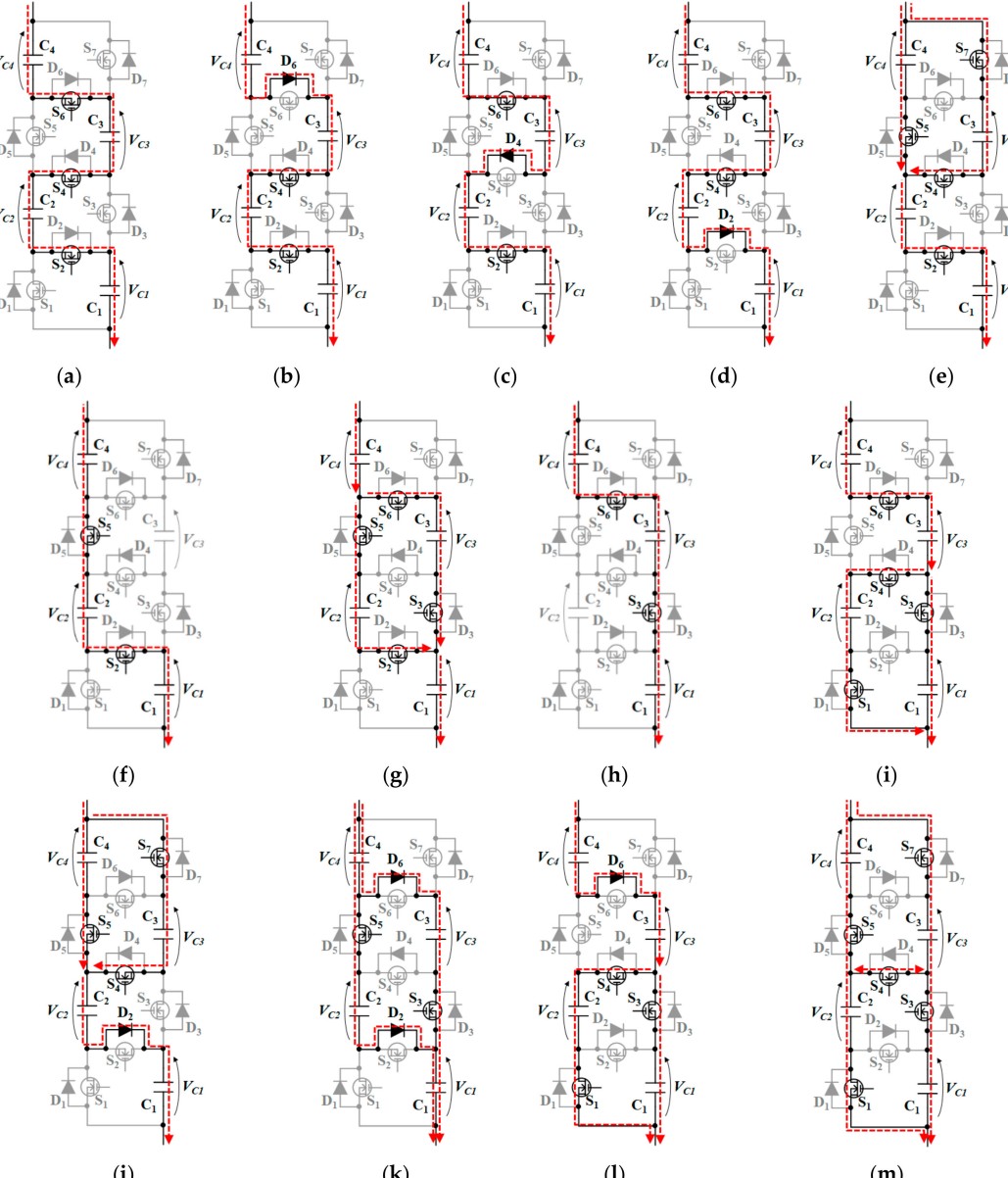

**Figure 12.** Mode transitions in charging process: current flow paths in (**a**) Mode 3, (**b**) dead time between Mode 3 and sub-Mode A, (**c**) dead time between Mode 3 and sub-Mode B, (**d**) dead time between Mode 3 and sub-Mode C, (**e**) sub-Mode A, (**f**) dead time between sub-Modes A and B, (**g**) sub-Mode B, (**h**) dead time between sub-Modes B and C, (**i**) Mode C, (**j**) dead time between sub-Mode A and Mode 1, (**k**) dead time between sub-Mode B and Mode 1, (**l**) dead time between sub-Mode C, and Mode 1 (**m**) Mode 1.

Sub-Mode A in Mode 2 (Figure 12e): $S_4$ and $S_7$ are turned off so as to connect $C_3$ and $C_4$ in parallel. Voltages of $C_3$ and $C_4$ are balanced thanks to the parallel connection.

In Mode 2, sub-Modes A–C are repeatedly changed so that all the cell voltages are balanced. Meanwhile, the module configuration is kept 3S even in the dead time periods. The sub-mode transitions in Mode 2 are repeated until $V_C$ and $V_{mod}$ reach $V_{M1}$ and $3V_{M1}$, respectively.

Dead Time between Sub-Mode A and Mode 1 (Figure 12j): $S_2$ is turned off while $S_4$, $S_5$, and $S_7$ are still on. The anti-parallel diode $D_2$ starts conducting instead of $S_2$.

Mode 1 (Figure 12m): $S_5$ and $S_7$ are turned on so as to connect $C_1$ and $C_2$ in parallel. The module is in 2S-2P configuration, and therefore, $V_{mod}$ drops to $2V_{M1}$ at the beginning of Mode 1. Voltages of

parallel-connected cells are balanced automatically, and all cell voltages are uniform unless there is huge capacitance mismatch.

### 3.4. Operation Conditions

The series-parallel connection in the proposed circuit must be reconfigured so that $V_{mod}$ be within the upper and lower voltage limits of $V_U$ and $V_L$. From the profile of $V_{mod}$ in Figure 9a, following sets of equations can be yielded:

$$V_U \geq \begin{cases} 2V_{cha} \ (t = 0) \\ 3V_{M1} \ (t = T_1) \\ 4V_{M2} \ (t = T_2) \end{cases} \tag{2}$$

$$V_L \leq \begin{cases} 2V_{M1} \ (t = T_1) \\ 3V_{M2} \ (t = T_2) \\ 4V_{M3} \ (t = T_3) \end{cases} \tag{3}$$

where $T_1$–$T_3$ are the time at the end of Modes 1–3 in the discharging mode. Rearrangement of Equations (2) and (3) produces:

$$\frac{V_U}{3} \geq V_{M1} \geq \frac{V_L}{2} \tag{4}$$

$$\frac{V_U}{4} \geq V_{M2} \geq \frac{V_L}{3} \tag{5}$$

$V_{M1}$ and $V_{M2}$ must be determined to fulfill Equations (4) and (5). The relationship between $V_U$ and $V_L$ can be yielded from Equations (4) and (5), as:

$$V_U \geq 1.5V_L \tag{6}$$

### 3.5. Reconfiguration Algorithm

Assuming all cell voltages are equalized to be $V_C$, the module configuration is determined based on $V_C$, as:

$$\begin{aligned} V_C \geq V_{M1} &\rightarrow Mode \ 1 \ (2S - 2P) \\ V_{M1} > V_C \geq V_{M2} &\rightarrow Mode \ 2 \ (3S) \\ V_{M2} > V_C &\rightarrow Mode \ 3 \ (4S - 1P) \end{aligned} \tag{7}$$

The flowchart of the reconfiguration sequence for the four-cell module is depicted in Figure 13. Series-parallel configuration or Modes 1–3 are switched based on the value of $V_C$. Meanwhile, sub-modes in Mode 2 are sequentially changed at a low frequency (e.g., 4 Hz in the experimental test, Section 4).

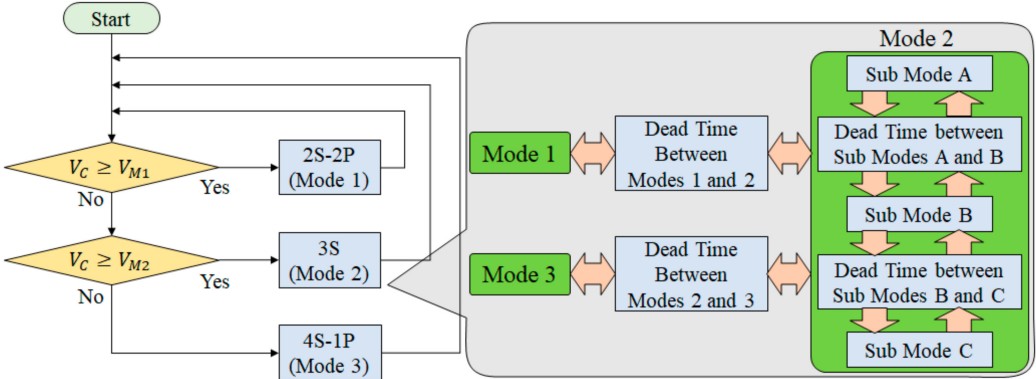

**Figure 13.** Flowchart of reconfiguration sequence.

In Modes 1 and 3, all cells are essentially charged/discharged uniformly as the series-parallel configurations of cells are fixed. In Mode 2, the series-parallel configuration is sequentially switched, as illustrated in Figures 11 and 12, and cell voltages are equalized by virtually connecting all cells in parallel. However, in each sub-mode, cell voltages slightly deviate due to nonuniform charge/discharge currents. For example, $C_1$ and $C_2$ in sub-Mode C (Figure 11i) are in parallel and are discharged uniformly, but a discharging current of $C_1$ in other sub-modes and dead time periods is twice that of $C_2$. Voltages of $C_1$ and $C_2$ are slightly imbalanced, and they are connected in parallel again when the operation comes back to sub-Mode C. This voltage imbalance might result in an excessively large current at the moment of the mode transition. Therefore, the voltage imbalance generated in Mode 2 should be low enough by changing sub-modes frequently.

A voltage imbalance between $C_1$ and $C_2$ generated in Mode 2 is focused here as an example, but a voltage imbalance between $C_3$ and $C_4$ can be analyzed identically. Sub-mode lengths as well as dead time periods are assumed equal to be $T_S$. From sub-Mode C as an origin, it takes $7T_S$ for the operation to come back to sub-Mode C again (see Figure 10). $C_1$ and $C_2$ are discharged with $I_{mod}$ and $I_{mod}/2$ ($I_{mod}$ being the module current), respectively, in sub-Modes A and B. The voltage imbalance generated before the operation comes back to sub-Mode C is given by:

$$\Delta V = \frac{7I_{mod}T_S}{2C} \tag{8}$$

where $C$ is the cell capacitance. This equation suggests $T_S$ should be properly determined depending on values of $I_{mod}$ and $C$ so that $\Delta V$ is low enough. In general, $I_{mod}/C$ for high-power applications (e.g., 20C rate) is around 0.014—$Q = CV = IT$ with $V = 2.5$ and $T = 180$ sec (=3600/20) yields $I/C \approx 0.014$. Assume $T_S = 0.25$ (i.e., 4 Hz), $\Delta V$ can be as low as 12 mV that is considered low enough to preclude excessive large current in parallel connection.

## 4. Design Example

A reconfiguration circuit for four cells, each with a capacitance of 400 F at a rated charging voltage $V_{cha} = 2.5$ V, with $I_{mod}$ of 0.4 A is assumed and key parameters of $V_U$, $V_L$, $V_{M1}$, and $V_{M2}$ are determined to achieve 90% utilization ratio. Although $V_U$, $V_L$, $V_{M1}$, and $V_{M2}$ can be an arbitrary value in a certain range, the lowest possible values are chosen in this article.

According to Equation (1) and Figure 2, $V_{cut}$ is determined to be 0.8 V to achieve 90% energy utilization, and hence, $V_{M3} = 0.8$ V. Applying $V_{cha} = 2.5$ V and $V_{M3} = 0.8$ V into Equations (2) and (3) yields $V_U \geq 5.0$ V and $V_L \geq 3.2$ V. The lowest values of $V_U = 5.0$ V and $V_L = 3.2$ V are chosen here, and these values satisfy Equation (6). Substitution of $V_U = 5.0$ V and $V_L = 3.2$ V into Equations (4) and (5) produces 1.66 V $\geq V_{M1} \geq$ 1.60 V and 1.25 V $\geq V_{M2} \geq$ 1.07 V, respectively. Accordingly, $V_{M1}$ and $V_{M2}$ are determined to be 1.60 V and 1.07 V, respectively.

## 5. Experimental Results

A prototype for a four-cell module was built and tested to verify the proposed reconfiguration concept. EDLCs with a capacitance of 400 F at a rated charge voltage of 2.5 V were used. The module was discharged and charged with a constant current of 0.4 A. $V_{M1}$ and $V_{M2}$ were set to be 1.60 V and 1.07 V, respectively, as determined in Section 4. An average cell voltage was used as $V_C$ for the reconfiguration control. The switching frequency in Mode 2 was 4 Hz.

The resultant discharging and charging profiles are shown in Figure 14. In the discharging mode, cells in Mode 1 were discharged with a slight voltage imbalance due to a minor initial voltage mismatch. When $V_C$ reached 1.60 V, the module was reconfigured to be 3S in Mode 2, and $V_{mod}$ rose. All cell voltages in Mode 2 were unified, and the initial voltage mismatch found in Mode 1 vanished thanks to the switched capacitor operation. The module was reconfigured to be 4S-1P when $V_C$ dropped to 1.07 V, and all cells were eventually discharged as low as 0.8 V at the end of Mode 3. In the charging

process (see Figure 14b), the module was reconfigured in reverse order. All cells were eventually charged to 2.5 V without causing voltage imbalance.

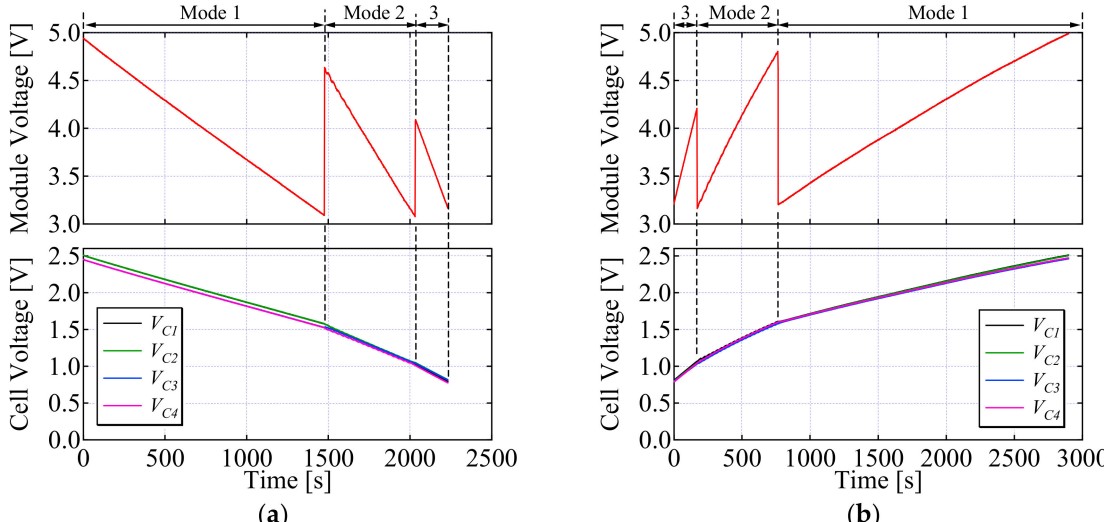

**Figure 14.** Experimental results. (**a**) Discharging profiles and (**b**) charging profiles.

The resultant discharging and charging profiles agreed very well with the theoretical ones shown in Figure 9, demonstrating the proposed reconfiguration technique. The module voltage variation was limited between 3.2 V and 5.0 V, while cells were cycled in the range of 0.8–2.5 V, achieving the energy utilization ratio of 90%.

Cell voltages in Mode 2 in the discharging mode are zoomed in Figure 15. All the cell voltages were equalized well, demonstrating the equalization performance of the proposed series-parallel reconfiguration circuit.

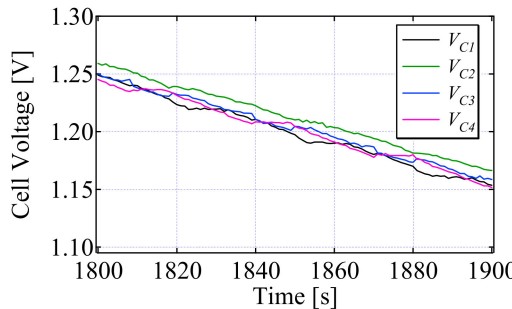

**Figure 15.** Zoomed cell voltage profiles in Mode 2 in discharging process.

## 6. Conclusions

The series-parallel reconfiguration circuit with voltage equalization capability has been proposed for EDLC modules in this paper. In comparison with conventional balance-shift circuits, not only can the switch count be reduced but also the proposed reconfiguration circuit can be applied to any arbitrary odd and even number of cells, improving the design flexibility and scalability. In addition, since the proposed circuit behaves as a switched capacitor converter, with which all cells are virtually connected in parallel, all cells can be charged and discharged uniformly, realizing the improved utilization ratio of the module compared to the conventional unbalance-shift circuits.

In this study, a prototype for an EDLC module comprising four cells, each with a rated charging voltage of 2.5 V, was built and tested. The experimental discharging and charging profiles agreed very well with theoretical ones, verifying the proposed concept. The module voltage in the experiment

varied between 3.2 V and 5.0 V while all cells in the module were cycled in the range of 0.8–2.5 V, achieving the energy utilization ratio of 90%.

**Author Contributions:** Conceptualization, M.U.; methodology, M.U.; simulation analysis, I.K. and H.K.; validation, I.K. and H.K.; writing—original draft preparation, M.U.; writing—review and editing, M.U.; supervision, M.U. and H.K.

**Funding:** This research received no external funding.

**Conflicts of Interest:** The authors declare no conflict of interest.

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
