# Peer review of "Series-Parallel Reconfiguration Technique with Voltage Equalization Capability for Electric Double-Layer Capacitor Modules"

_energies, doi:10.3390/en12142741_

Round 1
Reviewer 1 Report
Authors introduced a novel approach of series-parallel reconfiguration technique with voltage equalization capability for EDLC modules. The proposed technique can be applied to any number of cells.
Authors provided a solid state-of-the-art, explained thoroughly the principle of their technique during discharging, showed performance on a real application. The paper is very well-written, readable, and of minimum typos. The particular sections logically follow each other, the reader is well driven through the text. Accordign to my searches, the paper is original and inspiring. I recommend the paper for publications.
Yet I have some minor remarks
- Line 121, Authors state that "Diodes are connected in parallel with switches in order to provide current flow paths during dead time periods, as will be detailed in Section 3." However I would recommend to explain the function of diode D7.
- I would recommend to add figure (similar to Fig. 10) describing the charging process. It will increase the readability of the paper.
Author Response
Dear Editor and Reviewers
We are very grateful to you for your thoughtful and helpful review of the manuscript. Your comments and suggestions have been incorporated as appropriate into the revised manuscript. The revised parts are highlighted with green in the revised manuscript. The responses to reviewers’ comments are noted in the attached file.

Reviewer 2 Report
The paper presents a novel series-parallel reconfiguration technique with voltage equalization capability for EDLC modules. Compared to other techniques, the main advantages of this configuration are its potential scalability and its equalization capability.
Generally speaking, I think it is a very good paper: the main goals of the work are clearly stated, state-of-the-art is investigated adequately, the methodology is presented very clearly, also with the aid useful figures, English style was very good, and globally the paper is pleasant to read. The experimental validation is also a strong point of the article. However, some work must be done to improve further the quality of the paper.
1. The equalization capability should be deeply investigated and clearly explained. From the text, it is not possible detecting how the proposed reconfiguration technique is also able to equalize the cells.
2. Regarding Mode 2, during a submode, it is expected that the voltages of the cells in parallel will drift from the value of the other two cells. It is expected that switching from a submode to another submode will connect two cells with different voltage in parallel. How do you manage this? What could happen? The voltage evolutions over Mode 2 should also be investigated in the results.
3. Focusing on Vc, that is used for the control, which value of Vc is considered? Do you use an average value or do you calculate it from the measured value of the module voltage Vmod?
4. Regarding the experimental results, the charging and discharging process is clear, but much more experiments should be added. Especially, a power profile simulating real applications (fast power fluctuations, power flow inversions, etc.) could provide interesting information about the performances of the proposed configuration. Also balancing performance should be investigated better in more undesired operating conditions.
5. The authors should spend some words on an eventual configuration with an odd number of cells. In addition, since this configuration is exploitable with a higher number of cells, which are the configurations the authors think would be used? For example, what would happen with 6 cells, or 8 cells? Or a very high number, like more than 20? Do they think that the solution would be keeping the 4 cell configuration as a module to be replicated for each cluster of 4 cells?
6. The quality of the figures of experimental results should be enhanced. Please, also add a grid.
Author Response

(The authors gave the same response as above.)
